# Coparenting and Mental Health in Families with Jailed Parents

**DOI:** 10.3390/ijerph18168705

**Published:** 2021-08-18

**Authors:** Eman Tadros, Kerrie Fanning, Sarah Jensen, Julie Poehlmann-Tynan

**Affiliations:** 1Division of Psychology and Counseling, Governers State University, University Park, IL 60441, USA; 2Department of Human Development and Family Studies, University of Wisconsin–Madison, Madison, WI 53706, USA; kafanning@wisc.edu (K.F.), julie.poehlmanntynan@wisc.edu (J.P.-T.); 3Department of Sociology, University of Wisconsin–Madison, Madison, WI 53706, USA; spjensen3@wisc.edu

**Keywords:** actor–partner interdependence model, coparenting, family systems theory, incarcerated coparenting, jail, mental health, parental incarceration

## Abstract

The number of families affected by parental incarceration in the United States has increased dramatically in the past three decades, with primarily negative implications for adult mental health and child and family well-being. Despite research documenting increased strain on coparenting relationships, less is known regarding the relation between adult mental health and coparenting quality. This study investigated coparenting in families with young children currently experiencing parental incarceration. In a diverse sample of 86 jailed parent–caregiver dyads (*n* = 172), this analysis of a short-term longitudinal study examined the links among jailed parents’ and children’s at-home caregivers’ externalizing mental health symptoms and perceived coparenting alliance quality using the Actor–Partner Interdependence Model. Analyses using structural equation modeling revealed a medium sized negative partner effect for externalizing behaviors on coparenting alliance for jailed parents, wherein caregivers increased externalizing symptoms related to jailed parents’ lower reported coparenting quality. Caregiver–partner effects and both actor effects resulted in small effects. These findings highlight the roles of mental health and coparenting relationship quality when a parent is incarcerated and contribute to the existing literature on incarcerated coparenting, with implications for theory and practice.

## 1. Introduction

Although only five percent of the world’s population resides in the United States (U.S.), it is home to 25% of the world’s prison population [1], as the U.S. has the highest incarceration rate of any country [2]. Most incarceration in the U.S. occurs in jails as opposed to prisons [3], and the majority (67%) of individuals incarcerated in jails are parents [4]. Stark racial and economic disparities in incarceration rates result in more impoverished families and families of color impacted by parental incarceration [5]. Parental incarceration impacts families and individuals, including effects on the incarcerated parent’s mental health, the mental health of children’s at-home caregivers, the parent–caregiver coparenting relationship, and children’s well-being [6]. From a family–systems theory (FST) perspective [7], positive coparenting between incarcerated parents and their children’s at-home caregivers (henceforth referred to as caregivers) may act as a dyadic protective factor for the family system; however, few studies have examined the incarcerated coparenting relationship as a dyadic system. In the current study, using the Actor–Partner Interdependence Model (APIM) [8], we explore jailed parents and caregivers as dyads, including associations among incarcerated parent and caregiver externalizing mental health symptoms and their coparenting alliance. This study is novel in its dyadic analysis of the perspectives of the incarcerated and non-incarcerated coparents, as prior studies have investigated one side of the coparenting relationship or treated the perspectives independently.

### 1.1. Jailed Parents and Family Relationships

In the U.S., although prisons house more incarcerated individuals for extended periods of time, usually for felony crimes, local jails act as the front-door to incarceration. Jails cause much of the churn in and out of corrections facilities, partly because most individuals in jail have not been convicted or sentenced, whereas a minority are sentenced to jail for misdemeanor crimes [9]. Outside the walls of corrections, 13.5% of American adults have experienced a spouse’s or coparent’s incarceration [10] and more than five million U.S. children experience the incarceration of a co-resident parent [5].

Families of jailed parents often experience increased disruption stemming from the cyclical nature of jail incarceration [11]. In addition, parental incarceration is intimately linked with prior disadvantage (e.g., poverty, homelessness, substance problems) while simultaneously transmitting these disadvantages to their children in the future via criminal justice system contact [12]. Furthermore, families with an incarcerated loved one experience tremendous stigma surrounding the incarceration [13], potentially inhibiting family communication. The stigma can also impede access to social support networks, further straining the coparent relationship and relationships with children [14].

Due to their short-term stays, jails are less likely than prisons to facilitate programs to support family relationships and functioning [15]. Correctional facilities are even less likely to offer these programs to fathers than mothers because of gendered roles and parenting stereotypes [16]. Given that the majority of incarcerated parents are fathers [17], though rates of maternal incarceration have increased over recent years, especially in jails [18], the majority of parents in jail receive little to no family programming, even when mental health concerns are present. The strain on families and limited programs are important reasons to investigate incarcerated and non-incarcerated coparents as dyads in the family context.

### 1.2. Jailed Parents’ and Caregivers’ Mental Health

Incarcerated individuals frequently report mental health concerns, including depression, substance misuse, and alcoholism [19,20]. This is true for parents incarcerated in jails as well (e.g., [21,22]). In addition to elevated mental health symptoms in incarcerated parents, children’s at-home caregivers are also at risk for experiencing stress, diminished parenting quality, and elevated mental health symptoms during a parent’s incarceration [23]. For example, Wildeman et al. [24] reported that mothers experienced increased likelihood for major depressive episodes and decreased life satisfaction following a father’s incarceration. Caregiver mental health has implications for coparenting quality and relationships with children [25]. In a longitudinal study, Kinner et al. [26] found that much of the link between paternal incarceration and poor youth outcomes related to maternal anxiety, depression, and substance use, in addition to low socioeconomic status and low parental supervision. Furthermore, variation in children’s outcomes exist, related to children’s interactions with their residential caregiver and the caregiver’s mental health and stability [27]. Investigating coparenting in this context may be especially important given residential caregivers frequently report parenting fatigue and lack of parenting support [28,29], which may magnify caregiver mental health strain.

### 1.3. Incarcerated Coparenting

The coparenting alliance is a partnership between individuals parenting a child that involves coordination of care [30]. Incarcerated coparenting is a relationship whereby the incarcerated person “…(biological, foster or legal guardian) provides love, nurturance, and care while being involved with and/or held by the justice system while negotiating roles, rules, responsibilities, and contributions with a partner parent” [31] (p. 4). Incarcerated coparenting involves reliability of responsibility and decision cohesion within the coparenting alliance [32,33]. Research has shown that strengthening the alliance between coparents is linked with positive moods and interactions among family members [34]. Strong coparenting relationships can promote positive parent–child relationships [35] and enhance developmental outcomes in children [36]. While relations between coparenting quality and mental health have been evidenced by previous research in community settings (e.g., [37]), it is unknown how incarcerated parents’ and caregivers’ mental health relates to coparenting quality during parental incarceration, which may have implications for incarcerated parent–child relationship quality (e.g., [34]). Previous research has found that incarcerated parent–caregiver coparenting relates to parent–child contact during and after incarceration [38], reentry outcomes [29,39], and family well-being [40]. In addition to the challenges related to having an incarcerated partner or coparent, parents and caregivers must work together and agree on ways to resolve coparenting conflicts to facilitate the child’s well-being [41].

Coparenting relationship quality may be especially important during early childhood (birth–age 6) given the tremendous growth of lifelong skills and relationships that occur during this developmental stage [42]. Pudasainee-Kapri and Razza [43] found that during infancy, supportive coparenting is associated with greater father engagement at 1 year of age and higher levels of child to mother attachment at age 3. Additionally, Poehlmann [44] indicated that the majority of young children with incarcerated mothers in their sample exhibited representations of insecure attachment. Furthermore, coparenting during early compared to later childhood and adolescence presents a unique context for understanding coparenting relationships, for coparents are often beginning the process of negotiating roles, rules, and responsibilities with each other as young children rapidly grow, requiring frequent reestablishment of the coparenting alliance to align with the child’s growth (e.g., [45]).

Given these well-documented links between incarcerated parent–child relationship quality and short- and long-term parent, caregiver, and child well-being outcomes, positive coparenting alliances may serve as a protective factor for children experiencing parental incarceration. Indeed, making decisions around new boundaries, rules, and roles related to incarceration may foster family structure and cohesion [31]. Family Systems Theory (FST) [7] may help explain how incarcerated coparenting should be conceptualized as a dyadic unit.

### 1.4. Family Systems Theory

FST is a perspective in which family members’ interactions, patterns, and relationships are conceptualized by how they impact individuals, couples, and families as part of a system [7,46,47]. FST utilizes many concepts relevant to incarcerated coparenting, including holism, morphogenesis, morphostasis, and feedback. FST postulates that individuals can only be understood in the context of their relationships [48] rather than blaming any one individual for family problems. This is particularly relevant to the lives of incarcerated individuals who are often blamed and stigmatized for the issues present in the family system due to their incarceration.

Holism is the concept that explains the pieces that make up a system (e.g., biological, psychological, economic, social) and that “the whole is more than a sum of its parts” [49] (p. 18). For example, holism explains that we cannot understand complex family dynamics during parental incarceration by viewing each individual in a family as a separate entity. Therefore, APIM is an ideal model for studying the coparenting relationship as the partners can be viewed as a system rather than as individuals. Morphogenesis is the system’s capacity to change given new circumstances, while maintaining a level of communication and interaction [50], abilities that are essential for effective coparenting and that are directly impacted by parental incarceration (e.g., physical separation, limited communication avenues). Morphostasis is the system’s capacity to maintain stability despite new circumstances and generally sustain the system’s role in the larger macrosystem and each individual’s role within the family [50]. Both stability and change are needed to adjust to parental incarceration.

In addition, positive and negative feedback (i.e., explicit and implicit social messages) can come from both inside and outside the family, from other systems such as extended family, community, school, law enforcement, and other aspects of the larger macrosystem. For example, feedback from the criminal justice system (e.g., importance of parent–child visits, stigma, restorative perspectives to incarceration) can influence the family, depending on how open or closed the family system is. Further, the incarcerated parent and caregiver also affect each other and their relationships with children via positive and negative feedback. Because caregivers have daily access to children and incarcerated parents often rely on caregivers for information about and contact with their children, it is likely that caregivers are able to affect incarcerated parents via this feedback (more than the other way around).

In addition to the concepts above, it is critical to note that systems consist of boundaries that dictate the amount of information that is allowed to flow between members within the system and among other systems outside the family. These boundaries are ideally consistent, relatively flexible yet clear, and well-understood among members of the system [50]. In families affected by parental incarceration, the flow of information is often regulated by family members who are at home, including caregivers. When clear boundaries are not in place, dysfunction may occur [51,52]. With boundaries and other concepts, FST situates the family as a cohesive whole, rather than as individual actors. A dyadic analysis enables examination of the experiences of incarcerated and non-incarcerated coparents within the dyad, rather than as individuals, consistent with FST.

### 1.5. Current Study and Research Questions

The current study examines incarcerated parents and caregivers as dyads, exploring preliminary associations among mental health and coparenting from a family systems perspective. Coparenting is a dyadic process, in which neither member of the dyad is independent of the other. The relationship consists of two family members who influence and respond to each other. Thus, we utilized APIM [8], as it enables estimation of an effect of an independent variable on a dependent variable within a person (actor effect), as well as an independent variable’s effect on the partner’s dependent variable (partner effect), in paired dyads (see Figure 1). We aimed to address the following research question: How do jailed parent and caregiver externalizing mental health symptoms relate to perceived coparenting alliances? Given previous literature regarding coparenting and mental health in non-incarceration contexts, we first hypothesized that jailed parents’ and caregivers’ coparenting alliance ratings would be positively correlated, where when one member of the dyad scores high, the other member also scores high. Second, we hypothesized that there would be actor effects for both jailed parents and caregivers (Figure 1, pathways 1 and 2). Additionally, given the mutual influence of members presented in FST, we also anticipated partner effects (Figure 1, pathways 3 and 4).

## 2. Materials and Methods

The current study utilized a subsample of initial assessment data from a larger, short-term longitudinal mixed methods study on the sequelae of parental incarceration for young children (2–6 years, *M* = 4.1 years, *SD* = 1.3, 54.9% boys) and families. In the larger study, data were collected from incarcerated parents, randomly-selected focal children, and children’s caregivers regarding well-being, family relationships, parent–child and caregiver–child interactions, incarceration-related experiences, housing experiences, access to community supports, and incarcerated parent–child visits and contact since the incarceration. The larger study consisted of 165 incarcerated parents, with 86 at-home caregivers and children also participating. Families participated in an initial assessment, including interviews, standardized measures, and observations both at the jail with the parent and during a home visit with the caregiver and child. A subsample also completed an observation during an incarcerated parent–child visit at the jail and a follow-up interview regarding the visit. The present study focused on the 86 incarcerated parent–caregiver dyads who reported on their coparenting alliance with each other and their own mental health at initial assessment. For all participating families, the incarcerated parent was engaged or coresident prior to the incarceration.

### 2.1. Participants

Parents were between 18–49 years old (*M* = 29.1 years, *SD* = 5.83; 84.8% fathers, 15.2% mothers) and caregivers’ ages ranged from 18–62 years old (*M* = 31.3 years, *SD* = 10.0). The majority of caregivers were the child’s biological mother (80.2%), 12.8% were grandparents, and 4.7% were the child’s biological father. The remaining 2.3% of caregivers were other relatives (e.g., aunt, uncle) or non-relatives (e.g., foster parent). In the study, 44.8% of jailed parents identified as Black, 33.3% White, 7.3% Latinx, and 14.6% multiple or other races. Caregivers identified as 37.2% Black, 47.7% White, 4.7% Latinx, and 5.8% multiple or other races. Parents had between 1–10 children, with an average of two children, and the majority of incarcerated parents lived with their child(ren) prior to incarceration (73.5%). Randomly selected during the larger study, focal children ranged in age from 2 to 6, with a mean of 4 years. Caregivers reported caring for 2 children on average (mode = 1 child), ranging from 1–6 children. We chose to focus on parents of children at this age, because it is a common age to have an incarcerated parent [5] and because the parent–child separation is so important at this age [53]. Table 1 presents a summary of sample demographic characteristics.

Parents were incarcerated for drug-related charges (15%), probation violations (21%), battery/violence (13%), nonpayment of child support (15%), domestic dispute/domestic violence (17%), driving under the influence (DUI) or driving while impaired (DWI) (11%), and other crimes (e.g., theft, property damage, 8%). Seventy-four (89.2%) parents reported previous experiences of incarceration. A few parents had served time in prison prior to transferring to jail, thus sentence length varied widely. Parents’ sentences ranged from 10 days to nearly 5 years, with an average sentence of 7 months.

### 2.2. Procedure

Jailed parents were recruited from three midwestern jails, all of which are run by their local sheriff’s department and have significant racial disparities in incarceration. Specific details regarding the jails can be found in Poehlmann-Tynan et al.’s study [54]. Parents with children between 2 and 6 years of age were identified by jail administrative staff and invited to participate in the study by a research assistant. Incarcerated parents participated in a brief screening to determine study eligibility (see Poehlmann-Tynan et al. [54] for inclusion and exclusion criteria). If an eligible incarcerated parent had more than one child in the age range, one child was randomly selected for participation (termed “focal child”). Parents who agreed to participate completed informed consent for themselves and their child approved by our university Institutional Review Board (protocol #SE-2010-0812). In addition, a National Institutes of Health (NIH) Certificate of Confidentiality was used. Following consent, parents provided contact information for the target child’s caregiver and participated in semi-structured interviews and self-administered questionnaires. Caregivers were then contacted by the research team and invited to participate in the study. During a home visit, participating caregivers completed informed consent for themselves and for the child (to account for any instances when the jailed parent was not the legal guardian) and completed semi-structured interviews and self-administered measures. Focal children participated in assessments by trained researchers. Families were given $50 for the home visit and children were given books. Incarcerated parents could not be compensated because of facility regulations.

### 2.3. Measures

#### 2.3.1. Demographics

To collect general demographic data, semi-structured interviews with caregivers and jailed parents were conducted by trained researchers. Information gathered during interviews included age, race, gender, income, education, public assistance use, and current/prior employment. These covariates were examined because of the extensive literature linking race and educational outcomes with incarceration [55,56,57]. In addition, incarcerated parents reported information specific to their incarceration (e.g., nature of crime, length of incarceration, previous incarcerations and communication routines with child and caregiver). Interviews with caregivers asked about their experiences of caregiving, their relationship to the incarcerated parent, and the child.

#### 2.3.2. Externalizing Mental Health Symptoms

Jailed parents and caregivers completed the Adult Self Report (ASR) [58], a standardized self-report mental and behavioral health questionnaire for adults ages 18–59. The ASR asks questions about behavioral, social, and emotional problems, including mental health, substance use, and about strengths and adaptive functioning (e.g., employment, education, family relationships). Example questions include: “I cry a lot”, “I am mean to others.” Based on the past 6 months, jailed parents and caregivers independently rated each problem item as 0, 1, or 2 (from not true to very true). Scores reported in the 97th percentile or higher fall within the clinical range [58]. This study used the externalizing (rule breaking, aggression, and intrusive subscales) composite scores, and all analyses utilize continuous T-scores calculated based on age and gender. Previous studies show high internal consistency for the ASR [59], with Cronbach’s alpha of 0.89 for the externalizing score. In the present sample, Cronbach’s alpha was 0.90 and 0.89 for jailed parents and caregivers, respectively. The percentage of jailed parents scoring at or above the clinical level (97th percentile or higher) for the externalizing behavior composites was 10.8%, and the percentage of caregivers was 4.9%.

#### 2.3.3. Coparenting Relationship Quality

To measure perceptions of coparenting and coparenting relationship quality, parents and caregivers each completed the Parenting Alliance Measure (PAM) [60] in relation to each other. The PAM consists of 20 questions about the other person regarding perceptions of the coparent’s relationship with the child, parenting style, the relationship between caregivers, and alignment between parenting styles. The PAM has been used previously with incarcerated fathers and mothers with minor children, with word changes to fit the context (e.g., replacing “child’s other parent” with “child’s caregiver”) [34]. Example questions include: *the child’s caregiver tells me I am a good parent*, *the child’s parent and I are a good team*. Items are summed into a total score; higher scores indicated parenting alliances characterized by more respect, communication, and teamwork, with scores of 20 or higher falling within the normal range, 19–15 within marginal, 14–16 within problematic, and scores less than 5 falling within the dysfunctional range. In this sample, Cronbach’s alphas for the PAM total score for jailed parents and caregivers were 0.95 and 0.96, respectively.

### 2.4. Analysis Plan

APIM allows researchers to investigate dyadic phenomena as nested data, where each member exists within an interdependent dyad [8]. In addition to estimating an effect of externalizing mental health symptoms (independent variable) on perceptions of coparenting alliance quality (dependent variable), the APIM also considers the correlational effect between the independent and dependent variables, accounting for shared attributes between partners and effects not resulting from the predictor variables (i.e., error terms, denoted as E in Figure 1). We conducted our analyses using the APIM_SEM online software [61]. We included the incarcerated parents’ race as a binary-coded between-dyad covariate (reference category = white), and education of the caregiver and jailed parent as within-dyad covariates. To determine whether the age of the focal child additionally influenced the coparenting relationship quality, focal child age was included in a separate model as an additional between-dyad covariate. However, focal child age was not a significant predictor of coparenting relationship quality and the resulting model did not differ from the original, and thus focal child age was not retained as a covariate in the final model. Analyses use structural equation modeling with maximum likelihood estimation using the Lavaan interface with R [62]. There were minimal missing data, ranging from 0–6% of responses across variables. Full information maximum likelihood (FIML) was used to address missingness. FIML is a model-based method often used in Structural Equation Modeling (SEM), with parameters at the population level directly estimated based on all information in sample data and advantages over other methods [63]. Comparison of the models with and without FIML evidenced no substantive changes.

#### Distinguishability and Power Analysis

In dyadic data analyses, dyads can be treated as either distinguishable (actor and partner effects are allowed to vary) or indistinguishable (actor and partner effects are constrained to be equal). Choosing between distinguishable and indistinguishable models is based on both empirical and theoretical justifications [8]. Tests of distinguishability providing a Chi-square statistic are presented, where significant results suggest the distinguishable dyad model is better fitting. However, given that jailed parents and caregivers are theoretically distinguishable (i.e., present theoretically different roles within the dyad), we present results for the distinguishable models only for all three hypotheses as suggested by Kenny et al. [8]. For models with distinguishable dyads, standardized beta estimates (*β(o)*) utilize a grand mean and standard deviation (akin to a weighted average) across both roles rather than within each role separately. This allows for the standardized estimates to be compared across roles [8,61].

Power analyses provide evidence of the likelihood of correctly rejecting the null hypothesis when in reality it is false (1-*β*). For dyadic data, power analyses are corrected for the level of nonindependence of the data and the interdependence of the study variables, resulting in separate power analyses for each role and effect. Power analyses for this model indicated adequate power to detect jailed parent partner effects (1-*β* = 0.94), however, jailed parent actor (1-*β* = 0.17) and both caregiver effects (actor 1-*β* = 0.28, partner 1-*β* = 0.05) were underpowered to detect small effects. 

## 3. Results

Descriptive statistics and correlations are presented in Table 2. All tests of coefficients used z tests, and effect sizes for actor and partner effects are presented below as partial correlations (*pr*). Bivariate correlations indicated significant negative correlations between caregivers’ externalizing behaviors and jailed parents’ perceptions of the coparenting alliance. This indicates that as caregivers’ externalizing symptoms increased, jailed parents perceived lower coparenting quality. As hypothesized, a significant positive correlation emerged between jailed parents’ and caregivers’ perceptions of the coparenting alliance, suggesting that when one member of the dyad perceived the coparenting relationship quality to be positive, the other member also perceived it to be positive, and vice versa.

### Jailed Parent and Caregiver Externalizing Mental Health Symptoms and Coparenting Alliance 

This model analyzed the effect of externalizing symptoms on the perceived coparenting alliance (Figure 2). Model results, confidence intervals, and standardized beta estimates (*β(o)*) are presented in Table 3. The R^2^ was 0.132 for jailed parents and was 0.037 for caregivers. The test of distinguishability showed that participants could be distinguished statistically by their incarceration status, χ^2^(16) = 37.57, *p* < 0.001. There were no statistically significant differences in intercept values, meaning that there was no main effect of whether the individual was incarcerated or not. There was significant correspondence between reported coparenting alliance controlling for the predictor variables (*r* = 0.481, *p* < 0.001, 95% CI [0.32, 0.96]), indicating that as one member perceived the coparenting relationship quality positively, so did the other member.

There were no statistically significant actor effects for the incarcerated parent, *B* = −0.075, *p* = 0.632, 95% CI (−0.383, 0.233), or the caregiver, *B* = −0.283, *p* = 0.167, 95% CI (−0.685, 0.118), with small effect sizes (jailed parents *pr* = −0.11; caregivers −0.15), suggesting caregivers’ and jailed parents’ externalizing symptoms had little effect on perceptions of their coparenting relationship. However, there was a statistically significant partner effect of the caregiver’s externalizing mental health on the jailed parent’s perceived coparenting alliance (*B* = −0.514, *p* < 0.001, 95% CI (−0.815, −0.214)), with a medium effect size of *pr* = −0.36. Thus, increased caregiver externalizing symptoms related to incarcerated parents’ reports of lower coparenting quality. Conversely, the jailed parent’s externalizing symptoms did not have a statistically significant effect on the caregiver’s perceived parenting alliance (*B* = −0.106, *p* = 0.607, 95% CI (−0.510, 0.298)), with a very small effect of *pr* = −0.01. The covariates, race (jailed parent *B* = −1.314, *p* = 0.693; caregivers *B* = −4.591, *p* = 0.298) and educational attainment (jailed parents *B* = 1.595, *p* = 0.314, 95% CI (−1.513, 4.703); caregivers *B* = −0.672, *p* = 0.743, 95% CI (−4.684, 3.339)) were not statistically significant.

## 4. Discussion

In this study of coparenting in families with young children who were experiencing parental jail incarceration, we examined preliminary links among adult externalizing mental health symptoms and coparenting quality from a family system perspective. Analyses revealed a medium sized partner-effect from caregivers’ externalizing symptoms to incarcerated parents’ coparenting alliance perception, but not the other way around. Specifically, higher caregiver externalizing symptoms related to jailed parents’ view of the coparenting alliance as less optimal. These preliminary findings expand on the existing literature by focusing on the coparenting dyad, with implications for family theory and practice.

In the current study, when caregivers reported more externalizing symptoms such as inattention, impulsivity, acting out, or arguing, incarcerated parents felt that the coparenting relationship was less optimal. These results are consistent with prior research in families not experiencing incarceration (e.g., [64]) and likely reflect the caregivers’ willingness (or unwillingness) or ability (or inability) to engage positively with the incarcerated parent around parenting of young children. Placing these results with in FST, these behaviors may lessen coparenting quality and present feedback to other systems, therefore making morphogenesis (being able to grow and adapt to change while maintaining a healthy structure) difficult to maintain (morphostasis).

The partner-effect of caregivers’ externalizing mental health symptoms on jailed parents’ coparenting perceptions is unsurprising given previous scholarship on caregiver gatekeeping. Resident caregivers often serve as gatekeepers for non-resident parents’ access to children. For example, in studies of unmarried fathers who do not reside with the mothers of their young children, the father–child relationship quality depends on the quality of the mother–father relationship (e.g., [65]). Similar dynamics have been documented in families experiencing parental incarceration, with children’s caregivers functioning as gatekeepers (e.g., [29]). Gatekeeping can be a reflection of power in the coparenting relationship or the gatekeeper’s desire to protect the child from a perceived threat or potential harm [66]. The latter may be especially true for caregivers who are worried about their children coming into contact with the criminal justice system because a parent is incarcerated. Given these common dynamics in families with nonresident or incarcerated parents, it is not surprising that the only partner effects documented in this study were from the caregiver to the incarcerated parent. From a family systems perspective, the caregiver may be in a position to set certain boundaries and regulate the flow of positive and negative feedback to the incarcerated parent. Thus, it is important to examine parental incarceration on the basis of holism, including family subsystems as well as individual well-being.

In general, caregivers have many external factors that contribute to their perceived coparenting besides those related to the incarcerated parent, including stressors at home or work, child factors, and social or coparenting support, whether from new partners, family members, or friends. Correctional systems present barriers to incarcerated parents’ relationships with and support from their extended family members, friends, and community, and thus they are particularly dependent on caregivers and their gatekeeping role regarding coparenting and access to the child. This phenomenon suggests that the coparenting dynamic may contribute to the communication frequency and relational outcomes through the caregiver’s gatekeeper role [67]. Whereas caregivers are free to make use of social and coparenting supports and thus, less likely to have their coparenting affected by the mental health of the incarcerated parent, they are also responsible for day-to-day care of the child. Perhaps other variables that are more child-related affect their perception of the coparenting alliance rather than the incarcerated parents’ or their own mental health, an area to be investigated in future studies.

Given these findings, FST can be helpful in addressing behaviors, interactions, boundaries, and overall communication within the whole system, including coparenting dyads and, subsequently, parent–child relationships. FST differs from typical psychotherapy perspectives, because it conceptualizes the family as a unit rather than separate individuals. By conceptualizing the family as a whole—with its range of interdependent dyadic and triadic subsystems—rather than individuals with problems, marriage and family therapists (MFTs) can provide clients with insight into how their larger systems play a role in their own interactions and behaviors in more ways than previously identified in order to gain a broader perspective on the issues presenting in therapy. Morphostasis and morphogenesis can help MFTs to interpret a parents’ incarceration as a form of dysfunction placed on the family system. Thus, it is vital to assess how the system is able to communicate and interact while these changes occur.

### 4.1. Implications for Practice

The results from the current study indicate that where dyadic coparenting and externalizing mental health symptoms are concerned, caregivers seem to influence jailed parents more than jailed parents influence their children’s caregivers. A key implication of the study is that more intervention research needs to be conducted with caregivers in the community and that research on corrections-based family programs needs to explicitly involve children’s caregivers. Although family program options are limited in jails, the few existing studies inform areas for further intervention research. For example, one pilot intervention in the Cook County Jail aimed to help incarcerated mothers and residential caregivers commit to coparenting together despite potential difficulties due to past conflicts and relationship complexities [68]. They focused on enhancing the four dimensions of coparenting as identified by Van Egeren and Hawkins [41]: solidarity, support, reduced undermining, and shared coparenting. In an evaluation of this intervention, Gleeson et al. [68] found that coparenting dyads who completed the program often had higher family functioning, as opposed to dyads who did not complete the program. Similarly, Miller et al. [15] implemented and evaluated the use of an evidence-based parenting program to jailed mothers which was previously offered for caregivers of children with incarcerated parents and for substance-abusing parents of school-aged children. The original program was manualized and tailored to the specific needs of this population. While this curriculum does not focus on coparenting, incarcerated mothers indicated that increased discussion of coparenting with relatives—often their own mothers—would be of interest in future iterations of the course. Miller et al. [15] also discuss the challenges of implementing such a course in a jail setting, especially with high attrition rates. Both of these programs focus on mothers, conforming to traditional gender roles despite the much larger number of incarcerated fathers in American jails. Gender-responsive parenting and coparenting programs are needed for incarcerated fathers as well.

### 4.2. Limitations and Future Directions

This study has several limitations that should be considered when interpreting the findings. First, the sample is small and not population-based, limiting generalizability and power. While there was significant power regarding jailed parent partner effects, power analyses revealed the model was underpowered regarding jailed parent actor effects and both caregiver effects, and as such the results should be considered with caution. Future research should use larger sample sizes when assessing APIM in families with incarcerated parents. Nonetheless, this study provides an interesting starting point that future research can build. Future studies with sufficiently powered dyadic datasets within the context of parental incarceration should further address these questions to better flesh out this and similar models. Second, while the larger study was a short-term longitudinal study, the data utilized in the current analyses were only collected at baseline; thus, we were not able to examine change over time in coparenting relationships and relations to mental health. Third, the within-group design does not allow comparisons with families who did not experience parental incarceration. Finally, coparenting relationship quality was based on self-report; in the future, observational data (such as during visits) could be used to indicate coparenting quality and dynamics. 

Directions for future research include further exploring the relations between jailed parent and caregiver mental health for coparenting relationships and examining the implications of incarcerated coparenting for parent–child relationships and children’s well-being. In the current study, we emphasized externalizing mental health behaviors. However, future studies should investigate possible associations between internalizing behaviors and perception of coparenting relationships within this context. Additionally, studies should investigate links between mental health and coparenting specific behaviors (e.g., communication frequency, negotiation), such as through timeline follow-back methods, daily diaries, observational assessments, and ecological momentary assessment methods. Future research could also examine children’s outcomes in relation to incarcerated parent–caregiver coparenting relationships, including implications for children’s mental health, social competence, happiness, and educational outcomes in relation to incarcerated coparenting.

## 5. Conclusions

In summary, this study examines the associations between jailed parents and caregivers externalizing mental health symptoms and their perceptions of the quality of their coparenting relationship. The findings indicate that caregivers’ increased externalizing behavior symptoms relate to jailed parents’ perception of decreased coparenting relationship quality. These preliminary results suggest the need for intervention research and programming targeting caregivers’ experiences during parental incarceration and point to the need for further research into the impacts of parental incarceration on coparenting relationships regarding mental health and child and family well-being.

## Figures and Tables

**Figure 1 ijerph-18-08705-f001:**
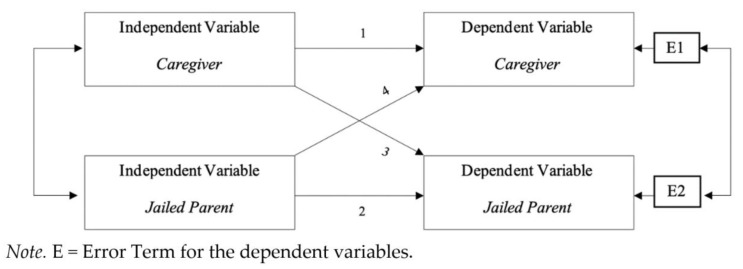
Actor and partner effects for jailed parents and at-home caregivers.

**Figure 2 ijerph-18-08705-f002:**
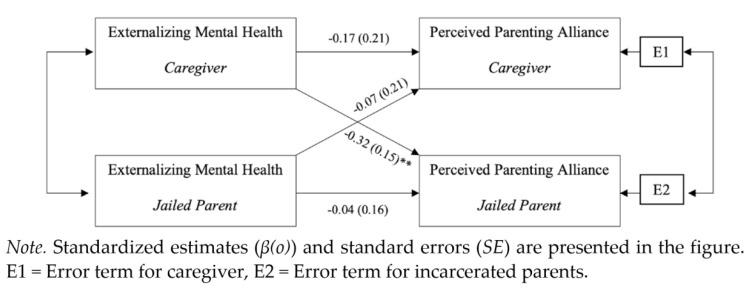
APIM results between externalizing symptoms and perceived coparenting alliance. ** *p* < 0.01.

**Table 1 ijerph-18-08705-t001:** Participant demographics.

Variable	Jailed Parent				Caregiver			
	*n*	%	Range	*M* ± *SD*	*n*	%	Range	*M* ± *SD*
Age	86	–	18–46	29.7 ± 6.2	86	–	18–62	31.3 ± 10.0
Relationship to Child								
Mother	9	10.5	–	–	69	80.2	–	–
Father	77	89.5	–	–	4	4.7	–	–
Grandmother	–	–	–	–	10	11.6	–	–
Other Relatives and Non-Relatives	–	–	–	–	3	3.6	–	–
Race								
African American	42	50.6	–	–	32	37.2	–	–
Caucasian	25	30.1	–	–	41	47.7	–	–
Latinx	5	6.0	–	–	4	4.7	–	–
Other or Multiple Races	11	13.3	–	–	9	10.5	–	–
Education								
Partial High School	18	21.7	–	–	17	19.8	–	–
High School Graduate	31	37.3	–	–	30	34.9	–	–
Partial College/College Graduate	34	41.0	–	–	39	45.4	–	–
Employment (% employed)	49	57.0	–	–	50	58.1	–	–
Income ($)	86	–	0–91,520	14,192 ± 13,849	86	–	0–140,000	15,357 ± 18,595

Note: Jailed parent employment and public assistance use are prior to incarceration. Race, education, and employment data were missing for 3 jailed parents. Income is presented in United States dollars ($).

**Table 2 ijerph-18-08705-t002:** Study variables means, standard deviations, and bivariate correlations.

Variable	*n*	*M*	*SD*	1.	2.	3.
1. Jailed Parent Externalizing	84	57.65	10.28			
2. Caregiver Externalizing	86	50.06	12.29	0.10		
3. Jailed Parent Perceived Coparenting Alliance	86	84.38	14.55	−0.10	−0.32 **	
4. Caregiver Perceived Coparenting Alliance	86	76.63	17.61	−0.02	−0.19	0.50 **

Note: Bivariate Pearson correlation coefficients (*r*) are presented. ^†^
*p* < 0.10; * *p* < 0.05; ** *p* < 0.01.

**Table 3 ijerph-18-08705-t003:** APIM analyses of jailed parent and caregiver externalizing behaviors and coparenting alliance.

Effect	Jailed Parent						Caregiver					
	*B*	*SE*	95% CI	*β(o)*	*pr*	1-*β*	*B*	*SE*	95% CI	*β(o)*	*pr*	1-*β*
Actor	−0.08	0.16	[−0.38, 0.23]	−0.04	−0.11	0.17	−0.28	0.21	[−0.68, 0.12]	−0.17	−0.15	0.28
Partner	−0.51 **	0.15	[−0.82, −0.21]	−0.32	−0.36	0.94	−0.11	0.21	[−0.51, 0.3]	−0.07	−0.01	0.05

Note: Controlling for jailed parent race (1 = white; 0 = non-white), jailed parent education (continuous), and caregiver education (continuous). *B* = Unstandardized estimate; *SE* = Standard error; 95% CI = 95% Confidence interval for the unstandardized estimate; *β(o)* = Standardized estimate using overall mean and standardized deviation for both roles; *pr* = partial r; 1-*β* = Power; * *p* < 0.05; ** *p* < 0.01.

## Data Availability

The data is being privately stored to protect the anonymity of the participants. A de-identified dataset is available upon request.

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
