# Peer review of "Coparenting and Mental Health in Families with Jailed Parents"

_ijerph, 2021, doi:10.3390/ijerph18168705_

Round 1
Reviewer 1 Report
This is really interesting study and provides really nice insight into the topic of FSM in parents who are incarcerated. The size of the sample was small which made it a little difficult to fully endorse the findings however this is a very hard sample to access and so this is justified. I really enjoyed reading this, the style of writing is easy and accessible and the authors did a great job in my view.
Some brief notes - The first use of APIM in the introduction is not defined correctly
In the method I expected to see more detail of larger study, reference to it even. the method states that the children are young- this may be something to consider in the article as an important impacting factor- being parent to younger children may also be a factor which impacts on relationships.
The size of the sample and the power are the concern but this has been stresses which is important
Author Response
Reviewer 1
Comments and Suggestions for Authors
This is really interesting study and provides really nice insight into the topic of FSM in parents who are incarcerated. The size of the sample was small which made it a little difficult to fully endorse the findings however this is a very hard sample to access and so this is justified. I really enjoyed reading this, the style of writing is easy and accessible and the authors did a great job in my view.
We thank reviewer 1 for their positive feedback and for acknowledging the barriers to recruitment within this context! We are glad to hear reviewer 1’s excitement regarding this study.
1. Some brief notes - The first use of APIM in the introduction is not defined correctly
Thank you for catching this mistake! We have corrected it in the manuscript on page 1.
2. In the method I expected to see more detail of larger study, reference to it even. the method states that the children are young- this may be something to consider in the article as an important impacting factor- being parent to younger children may also be a factor which impacts on relationships.
Thank you for this feedback. In the methods section, we have included more description of the larger study. If there is other information that we could include, we are happy to do so. Specifically, beginning on page 2, we have included:
“The current study utilizes a subsample of initial assessment data from a larger, short-term longitudinal mixed methods study on the sequelae of parental incarceration for young children (2-6 years, M=4.1 years, SD=1.3, 54.9% boys) and families. In the larger study, data were collected from incarcerated parents, randomly selected focal children, and children’s caregivers regarding well-being, family relationships, parent-child and caregiver-child interactions, incarceration-related experiences, housing experiences, access to community supports, and incarcerated parent-child visits and contact since the incarceration. The larger study consists of 165 incarcerated parents, with 86 at-home caregivers and children also participating. Families participated in an initial assessment, including interviews, standardized measures, and observations both at the jail with the parent and during a home visit with the caregiver and child. A subsample also completed an observation during an incarcerated parent-child visit at the jail and a follow-up interview regarding the visit. The present study focuses on the 86 incarcerated parent-caregiver dyads who reported on their coparenting alliance with each other and their own mental health at initial assessment. For all participating families, the incarcerated parent was engaged or coresident prior to the incarceration.”
We agree that the developmental age of the children (2-6 years) may present specific influences within this context. In our literature review section regarding coparenting, we have added a couple of sentences specific to coparenting young children. The following has been included on page 3:
“Coparenting relationship quality may be especially important during early child-hood (birth - age 6) given the tremendous growth of lifelong skills and relationships that occur during this developmental stage [42]. Indeed, Pudasainee-Kapri & Razza [43] found that during infancy, supportive coparenting is associated with greater father engagement at 1 year of age and higher levels of child to mother attachment at age 3. Additionally, Poehlmann [44] indicated that the majority of young children with incarcerated mothers in their sample exhibited representations of insecure attachment. Furthermore, coparenting during early compared to later childhood and adolescence presents a unique context for understanding coparenting relationships, for coparents are often beginning the process of negotiating roles, rules, and responsibilities with each other as young children rapidly grow, requiring frequent reestablishment of the coparenting alliance to align with the child’s growth (e.g., [45]).”
To address whether focal child age influenced the model, we re-ran the model including focal child age as a between-dyad covariate. Focal child age was not significant and the resulting model yielded similar results to the original. Because APIM is a saturated model, we are not able to compute fit statistics for the models to compare whether including focal child age added to the model’s fit. Due to this, we decided to leave focal child age out of the model. We have indicated this in section 2.4. On page 7, we have included the following:
“To determine whether the age of the focal child additionally influenced the coparenting relationship quality, focal child age was included in a separate model as an additional between-dyad covariate. However, focal child age was not a significant predictor of coparenting relationship quality and the resulting model did not differ from the original, and thus focal child age was not retained as a covariate in the final model.”
3. The size of the sample and the power are the concern but this has been stresses which is important
Thank you to reviewer 1 for your understanding regarding the limitations of this work. We recognize that the small study sample is a significant limitation of this work. We hope that we have clearly underscored this limitation, and welcome further feedback on it.
Reviewer 2 Report
Manuscript Number ijerph-1216492
Title: “Coparenting and mental health in families with jailed parents”
Present manuscript investigates mental health in families with jailed parents in United States, where it is home to 25% of the world’s prison population but only about 5% of whole world population. Sample was 86 jailed parent-caregiver dyads (N =172). Jailed fathers were 77, 89.5%, and jailed mothers, 9. Mental health was captured with measures of externalizing mental health symptoms, this study applies the adult self-report questionnaire (sample items, e.g., “I cry a lot”, “I am mean to others.”). Co-parenting relationship quality was captured with the parenting alliance measure (authors should indicate some sample items as in the other scale). Main results indicated (Table 2) a positive relation (r = .50) between the jailed parent and the caregiver in the co-parenting alliance (there are agreement between jailed parent and caregiver in the co-parenting alliance measure), and a negative relation between caregiver externalizing problems and jailed coparenting alliance (r = -.32). Last relation implicates that high jailed coparenting alliance are related with low caregiver externalizing problems. Additionally, low jailed coparenting alliance are related with high caregiver externalizing problems. This same relation seem also that was represented in Figure 2 (beta = -0.32, p<.01).
Parenting studies analyze the relation between parents and their children, or between parents that have to care their children. I believe that the study would be clearer if it were titled as for example: “One jailed parent: Parenting alliance and parents mental health”. Parenting alliance is about both parent’s relation (or any caregivers that act as one parent), but mental health is not expected that be the same for all family members.
The following 3 main questions. Is only related parenting alliance with parent’s mental health when is one jailed parent? Is expected that all family members, or only both parents, have the same mental health? Is expected any agreement between the both parents measure of their alliance?
Is possible when if you found no significative differences, this relation was not expected that be significative (you could be just in your confidence interval, i.e., 95%). It would be much better if you could calculate the sample size and power before doing the study, in the method section (for example, Fuentes et al., 2020; Garcia et al., 2021). Another issue to consider is that you will encounter a ceiling effect because of the population you work with in mental health measures.
References
Fuentes, M. C., Garcia, O. F., & Garcia, F. (2020). Protective and risk factors for adolescent substance use in Spain: Self-esteem and other indicators of personal well-being and ill-being. Sustainability, 12(5967), 1-17. doi:10.3390/su12155962
Garcia, O. F., Fuentes, M. C., Gracia, E., Serra, E., & Garcia, F. (2020). Parenting warmth and strictness across three generations: Parenting styles and psychosocial adjustment. International Journal of Environmental Research and Public Health, 17(7487), 1-18. doi:10.3390/ijerph17207487
Author Response
Reviewer 2
Comments and Suggestions for Authors
Present manuscript investigates mental health in families with jailed parents in United States, where it is home to 25% of the world’s prison population but only about 5% of whole world population. Sample was 86 jailed parent-caregiver dyads (N =172). Jailed fathers were 77, 89.5%, and jailed mothers, 9. Mental health was captured with measures of externalizing mental health symptoms, this study applies the adult self-report questionnaire (sample items, e.g., “I cry a lot”, “I am mean to others.”). Co-parenting relationship quality was captured with the parenting alliance measure (authors should indicate some sample items as in the other scale). Main results indicated (Table 2) a positive relation (r = .50) between the jailed parent and the caregiver in the co-parenting alliance (there are agreement between jailed parent and caregiver in the co-parenting alliance measure), and a negative relation between caregiver externalizing problems and jailed coparenting alliance (r = -.32). Last relation implicates that high jailed coparenting alliance are related with low caregiver externalizing problems. Additionally, low jailed coparenting alliance are related with high caregiver externalizing problems. This same relation seem also that was represented in Figure 2 (beta = -0.32, p<.01).
Thank you to reviewer 2 for their thoughtful and thorough review. We appreciate their feedback and welcome further suggestions for strengthening the paper.
1. Parenting studies analyze the relation between parents and their children, or between parents that have to care their children. I believe that the study would be clearer if it were titled as for example: “One jailed parent: Parenting alliance and parents mental health”.
We appreciate the recommendation to revise the title. However, after thorough consideration, we have determined that changing the title as suggested would misrepresent the sample and scope of the study for a portion of caregivers included in the sample also reported experiences of previous incarceration.
2. Parenting alliance is about both parent’s relation (or any caregivers that act as one parent), but mental health is not expected that be the same for all family members. The following 3 main questions.
-Is only related parenting alliance with parent’s mental health when is one jailed parent?
We appreciate the reviewer asking if parenting alliance matters even when someone isn’t incarcerated. In community settings, coparenting between two caregiving adults (often, but not always, the child’s biological parents) has been shown to be impacted by mental health (e.g., Williams, 2018). However, very limited research exists regarding coparenting and mental health when one member of the dyad is incarcerated. For the purposes of this paper we are focusing on coparenting when at least one member of the dyad is currently incarcerated. We have selected some recent articles below that also study incarcerated coparenting and have now cited them in the incarcerated coparenting section. This paragraph now reads, beginning on page 2:
“The coparenting alliance is a partnership between individuals parenting a child that involves coordination of care [30]. Incarcerated coparenting is a relationship whereby the incarcerated person “…(biological, foster or legal guardian) provides love, nurturance, and care while being involved with and/or held by the justice system while negotiating roles, rules, responsibilities, and contributions with a partner parent” [31] (p.4). Incarcerated coparenting involves reliability of responsibility and decision cohesion within the coparenting alliance [32-33]. Research has shown that strengthening the alliance between coparents is linked with positive moods and interactions among family members [34]. Strong coparenting relationships can promote positive parent-child relationships [35] and enhance developmental outcomes in children [36]. While relations between coparenting quality and mental health have been evidenced by previous research in community settings (e.g., [37]), it is unknown how incarcerated parents’ and caregivers’ mental health relates to coparenting quality during parental incarceration, which may have implications for incarcerated parent-child relationship quality [e.g., 34]. Previous research has found that incarcerated parent-caregiver coparenting relates to parent-child contact during and after incarceration [38], reentry outcomes [29, 39], and family well-being [40]. In addition to the challenges related to having an incarcerated partner or coparent, parents and caregivers must work together and agree on ways to resolve coparenting conflicts to facilitate the child’s well-being [41].”
Tadros, E. & Durante, K. (2021). Coparenting, negative educational outcomes, and familial instability in justice-involved families. International Journal of Offender Therapy and Comparative Criminology. https://doi.org/10.1177/0306624X211013740
​​Tadros, E., Durante, K., McKay, T., & Hollie, B. (2021). Coparenting from prison: An examination of incarcerated fathers’ consensus of coparenting. American Journal of Family Therapy. https://doi.org/10.1080/01926187.2021.1913669
​​Tadros, E. & Ogden, T. E. (2020). Conceptualizing incarcerated coparenting through a structural family theory lens. Marriage & Family Review, 56(6), 535-552. https://doi.org/10.1080/01494929.2020.1728007
Williams, D. (2018). Parental depression and cooperative coparenting: A longitudinal and dyadic approach. Family Relations, 67(2), 253–269. https://doi.org/10.1111/fare.12308
3. Is expected that all family members, or only both parents, have the same mental health?
Thank you for this comment. This paper focuses solely on the jailed parent-caregiver dyad as coparents, and subsequently does not include other family members within the analyses. To help clarify this, we have added the definition of incarcerated coparenting that we utilize for the current study. Specifically, on page 2, we have included: “Incarcerated coparenting is a relationship whereby the incarcerated person “…(biological, foster or legal guardian) provides love, nurturance, and care while being involved with and/or held by the justice system while negotiating roles, rules, responsibilities, and contributions with a partner parent” [31] (p.4).” While we do have information regarding child outcomes (e.g., child mental health, family relationships), we felt that adding more analyses to this paper was beyond the scope of this specific paper, and thus are not able to answer your question regarding other family members’ (e.g., children) mental health in relation to the current analyses. In the future we hope to continue this work to look at intergenerational relations between the coparenting dyad and the children. We welcome further feedback from the reviewer regarding this.
4. Is expected any agreement between the both parents measure of their alliance?
Thank you for raising this question. Given previous literature, we expected jailed parents and caregivers scores on the coparenting alliance measure to be similar, representing a significant positive correlation. We have added this to our hypotheses, which now reads (page 4): “Given previous literature regarding coparenting and mental health in non-incarceration contexts, we first hypothesize that jailed parents’ and caregivers’ coparenting alliance ratings would be positively correlated, where when one member of the dyad scores high the other member also scores high. Second, we hypothesize that there will be actor effects for both jailed parents and caregivers (Figure 1, pathways 1 and 2). Additionally, given the mutual influence of members presented in FST, we also anticipate partner effects (Figure 1, pathways 3 and 4).”
Additionally, we have included example questions for the Coparenting Alliance Measure as requested (page 7): “Example questions include: the child’s caregiver tells me I am a good parent, the child’s parent and I are a good team.”
5. Is possible when if you found no significative differences, this relation was not expected that be significative (you could be just in your confidence interval, i.e., 95%). It would be much better if you could calculate the sample size and power before doing the study, in the method section (for example, Fuentes et al., 2020; Garcia et al., 2021).
We thank reviewer 2 for this recommendation. As exemplified in the suggested citations, we have moved our discussion of the power analyses (originally section 3.1) to the method section (now section 2.4.1). The placement of Table 3 has been slightly adjusted to accommodate this change (now presented after section 3.1). Specifically, on page 8, it now reads:
“Power analyses provide evidence of the likelihood of correctly rejecting the null hypothesis when in reality it is false (1- β). For dyadic data, power analyses are corrected for the level of nonindependence of the data and the interdependence of the study variables, resulting in separate power analyses for each role and effect. Power analyses for this model indicated adequate power to detect jailed parent partner effects (1- β= .94), however, jailed parent actor (1- β= .17) and both caregiver effects (actor 1- β= .28, partner 1- β= .05) were underpowered to detect small effects.”
6. Another issue to consider is that you will encounter a ceiling effect because of the population you work with in mental health measures.
We thank reviewer 2 for this important critique. Individuals with mental illnesses are over-represented in the American criminal justice system and incarcerated population, and we would expect that the mean and median mental health scores for our sample to be higher overall for the incarcerated parent group compared to the residential caregiver group and the overall American, norm referenced population. However, this trend is representative of the larger incarcerated population, and it is not unique to our study. In response to this concern, we have included reports of the percentage of jailed parents and caregivers whose scores fell at or above the 97th percentile. In doing so, we found that 10.8% of incarcerated parents and 4.9% of caregivers reach this percentile of externalizing behavior. This suggests that a ceiling effect in our analyses is not likely. We have clarified this in our paper on Page 7: “The percentage of jailed parents scoring at or above the clinical level (97th percentile or higher) for the externalizing behavior composites was 10.8%, and the percentage of caregivers was 4.9%.”